# Investigation of Intra-Nitride Charge Migration Suppression in SONOS Flash Memory

**DOI:** 10.3390/mi10060356

**Published:** 2019-05-29

**Authors:** Seung-Dong Yang, Jun-Kyo Jung, Jae-Gab Lim, Seong-gye Park, Hi-Deok Lee, Ga-Won Lee

**Affiliations:** 1Department of Electronics Engineering, Chungnam National University, Daejeon 305-764, Korea; sdyang83@gmail.com (S.-D.Y.); jjk1006@cnu.ac.kr (J.-K.J.); jaegabi@cnu.ac.kr (J.-G.L.); hdlee@cnu.ac.kr (H.-D.L.); 2SK Hynix Inc., Gyeongchung-daero, Bubal-eub, Icheon-si 17336, Korea; pskg@sk.com

**Keywords:** SONOS, flash memory, charge spreading, plasma treatment, Oxygen-related trap, data retention

## Abstract

In order to suppress the intra-nitride charge spreading in 3D Silicon-Oxide-Nitride-Oxide-Silicon (SONOS) flash memory where the charge trapping layer silicon nitride is shared along the cell string, N_2_ plasma treated on the silicon nitride is proposed. Experimental results show that the charge loss decreased in the plasma treated device after baking at 300 °C for 2 h. To extract trap density according to the location in the trapping layer, capacitance-voltage analysis was used and N_2_ plasma treatment was shown to be effective to restrain the interface trap formation between blocking oxide and silicon nitride. Moreover, from X-ray Photoelectron Spectroscopy, the reduction of Si-O-N bonding was observed.

## 1. Introduction 

The NAND flash memory market is continuously growing by the successive introduction of mass data storage applications in portable electronic devices, such as USB memory and solid-state drives for tablet PCs and laptops [1]. The cell price as well as bit density are key factors in this application. Until now, it has been possible to reduce the bit cost and increase the bit density through the linear scaling down of cell size, which has been achieved by advanced lithography [2]. Recently, however, the NAND Flash memory industry has faced a scaling limitation of the conventional floating gate (FG) NAND cell. In order to find an alternative technology, Silicon-Oxide-Nitride-Oxide-Silicon (SONOS) device has received attention from researchers, as it provides simpler process steps, lower cell to cell coupling, and virtual immunity to stress-induced leakage current (SILC), when compared to FG [3,4,5]. However, the down-scaling process is still challenging in SONOS when attempted beyond the 30nm generation. To overcome the problem, SONOS has been fabricated with 3-dimesional (3D) structures such as BiCS [6], P-BiCS [7], TCAT [8], VG-NAND [9] and SMArT [10]. However, in the 3D SONOS structure, the charge trapping layer is not isolated but shared in a cell string, as shown in Figure 1. Due to this continuous trapping layer structure in the 3D scheme, the intra-nitride charge spreading can be a serious problem for data retention properties [11,12]. Charge spreading in silicon nitride has previously been studied in NROM devices, where a trapped charge is locally distributed, and recent research has reported that charge spreading is driven by the spatial concentration difference [13,14]. Figure 2a shows the probable charge spreading mechanism in silicon nitride. For trapped charges in deep-level sites, hopping can happen, yet the possibility is very low because of the long distance between deep-level sites. In the case of shallow-level sites, however, the hopping possibility increases due to relatively high concentration of trapping sites. Charge spreading via the shallow trap sites can be accelerated by conduction band diffusion of thermionic emitted carriers from the trap sites. Figure 2b shows trap energy levels in silicon nitride, and we can see that substitutional oxygen atoms at nitrogen vacancy causes a shallow-level trap site. Considering that the oxygen incorporation is active near the oxide/nitride interface, it is reasonable to estimate that the oxygen and nitrogen vacancy related defects will be formed near the nitride/oxide interface and that they are mainly located in shallow energy level, as reported in [15,16,17]. Figure 3 shows comparison results on the total number of bulk (N_Bulk_) and interface traps (N_int_) according to the channel radius of a cylinder type 3D SONOS device. Assuming N_Bulk_ = 1.0 × 10^18^ cm^−3^, relative importance of N_int_ increases as the channel radius decreases. Therefore, when the energy level of N_int_ is shallow, like as the oxygen related traps, the charge spreading via the interface trap sites becomes more critical with shrinkage of device dimension occurring.

In this study, N_2_ plasma treatment on silicon nitride is proposed to suppress the intra-nitride charge spreading by controlling the interface trap formation. To extract the trap density quantitatively, the capacitance-voltage (C–V) analysis was made based on the measurement results by a LCR meter (HP 4284A, Agilent, Santa Clara, CA, USA) at a small signal frequency of 1 MHz. To find the bonding state changes induced by plasma treatment, X-ray Photoelectron Spectroscopy (XPS) was also measured with a K-Alpha+ spectrometer (ThermoFisher Scientific, East Grinstead, UK).

## 2. Experiments

To fabricate SONOS structure, 6 nm SiO_2_ for tunneling oxide was thermally grown on a prime grade p-type Si substrate with high-purity oxygen gas via dry oxidation furnace. After the oxidation of Si, N_2_ plasma treatment was carried out for 30 sec. The flow rate of nitrogen gas was 45 sccm at a pressure of 10 mTorr, and a plasma power of 200 W. Silicon nitride as a charge storage layer was deposited by low-pressure chemical vapor deposition (LPCVD) at 825 °C with a gas flow rate of SiH_2_Cl_2_:NH_3_ = 170:70 sccm on the tunneling oxide. In this experiment, the nitride thickness varied between 7 nm, 15 nm, and 20 nm to extract the trap density by C–V analysis. Following this, N_2_ plasma treatment was performed once again on the top of nitride. Then, blocking oxide of 10nm was deposited by LPCVD at 680 °C, and 100 nm titanium (Ti) was deposited by RF-sputter for gate electrode. The test devices have a gate width by length of 100/100 μm. In order to investigate the impact of lateral charge migration on data retention, different gate stack structures were fabricated using a lithography mask, as shown in Figure 4. In extended structure (Ext. 10), the charge-trapping layer was extended to 10 µm in every direction of the gate electrode. In Ext. 10 structure, the gate etch was stopped on the blocking oxide layer, while the charge trapping layer was etched self-aligned with the gate in the reference devices (Ref.).

## 3. Results and Discussion

The program and retention behavior of the fabricated devices with and without N_2_ plasma treatment were measured as shown in Figure 5 and the charge loss during retention mode were calculated and are summarized in Table 1. The devices with extended trapping layer showed a larger memory window than the reference device, regardless of N_2_ plasma treatment. The reason for this is thought to be due to the fringe field effect of the extended devices. Furthermore, the over-etching issue has been shown to occur during the wet etching process in the reference devices, which in turn lowers program efficiency. However, the charge loss was larger after baking at 300 °C for 2 h. implying the intra-nitride charge spreading effect. The lateral charge loss of the extended devices was estimated to be about 28% in total charge loss. After N_2_ plasma treatment, the amount of charge loss deceased in the extended devices and the portion of lateral charge loss was 16%. For the quantitative comparison, nitride/oxide interface trap density was extracted using C–V method. When the positive bias was forced to the gate during C–V measurement, the charge was injected from the substrate and the flatband voltage (V_FB_) shifted due to charges captured at the traps. The V_FB_ shift (ΔV_FB_) enlarged with the increase in the ratio of occupied traps, and was finally saturated when all the traps were occupied. From the saturated ΔV_FB_, according to the trapping layer thickness as shown in Figure 6, a respective trap density of the silicon nitride can be calculated based on the formula below [18].
(1)$$\Delta V_{\mathrm{FB}}=\frac{qN_{\mathrm{BO}/\mathrm{TL}}}{\varepsilon_{{\mathrm{SiO}}_{2}}\varepsilon_{0}}T_{BO}+\frac{q}{\varepsilon_{\mathrm{SiN}}\varepsilon_{0}}\int_{0}^{T_{\mathrm{TL}}}xN_{\mathrm{Bulk}}(x)dx\;+\frac{qT_{\mathrm{BO}}}{\varepsilon_{{\mathrm{SiO}}_{2}}\varepsilon_{0}}\int_{0}^{T_{\mathrm{TL}}}N_{\mathrm{Bulk}}(x)dx+\left(\frac{T_{\mathrm{TL}}}{\varepsilon_{\mathrm{SiN}}\varepsilon_{0}}+\frac{T_{\mathrm{BO}}}{\varepsilon_{{\mathrm{SiO}}_{2}}\varepsilon_{0}}\right)qN_{\mathrm{BO}/\mathrm{TL}}\;=\frac{qN_{\mathrm{Bulk}}}{2\varepsilon_{\mathrm{SiN}}\varepsilon_{0}}T_{\mathrm{TL}}{}^{2}+\left(\frac{qT_{\mathrm{BO}}N_{\mathrm{Bulk}}}{\varepsilon_{{\mathrm{SiO}}_{2}}\varepsilon_{0}}+\frac{qN_{\mathrm{BO}/\mathrm{TL}}}{\varepsilon_{\mathrm{SiN}}\varepsilon_{0}}\right)T_{\mathrm{TL}}+\frac{qT_{\mathrm{BO}}N_{\mathrm{BO}/\mathrm{TL}}}{\varepsilon_{{\mathrm{SiO}}_{2}}\varepsilon_{0}}+\frac{qT_{\mathrm{BO}}N_{\mathrm{TO}/\mathrm{TL}}}{\varepsilon_{{\mathrm{SiO}}_{2}}\varepsilon_{0}}$$
where T_BO_, T_TL_, and T_TO_ are the thickness of blocking oxide, trapping layer and tunneling oxide. N_Bulk_ (cm^−3^) is the trap density of trapping layer and N_BO/TL_ (cm^−2^) and N_TO/TL_ (cm^−2^) are the interface trap density of blocking oxide/trapping layer and tunneling oxide/trapping layer, respectively, as shown in inset of Figure 6. From the dependency of ΔV_FB_ on the trapping layer thickness, N_Bulk_ can be assumed to be negligible and then, Equation (1) is expressed as follows.
(2)$$\Delta V_{\mathrm{FB}}=\frac{qT_{\mathrm{TL}}N_{\mathrm{BO}/\mathrm{TL}}}{\varepsilon_{\mathrm{SiN}}\varepsilon_{0}}+\frac{qT_{\mathrm{BO}}N_{\mathrm{BO}/\mathrm{TL}}}{\varepsilon_{{\mathrm{SiO}}_{2}}\varepsilon_{0}}+\frac{qT_{\mathrm{BO}}N_{\mathrm{TO}/\mathrm{TL}}}{\varepsilon_{{\mathrm{SiO}}_{2}}\varepsilon_{0}}$$

Based on Equation (2), the extracted trap densities are summarized in Table 1.

We can see that there was a distinct interface trap reduction in N_2_ plasma treatment, especially at blocking oxide and trapping layer (BO/TL) interface. Thus, charge loss decreased by 5.2% in extended N_2_ plasma devices. In the tunneling oxide and trapping layer (TO/TL) interface, the additional nitrogen supply effect by N_2_ plasma was ambiguous, but this may be because the nitrogen contributed to Si-O-N bonding formation on tunneling oxide, rather than curing the N vacancy in the nitride as the nitride was deposited after oxide formation. More consideration is needed to evaluate the accurate nitrogen behavior according to the underlying layer, but the results show that N_2_ plasma treatment was effective in reducing the interface trap between blocking oxide and silicon nitride while maintaining the nitride bulk trap.

For the physical analysis on N_2_ plasma effect, XPS was also measured on the oxide/nitride interface to find the bonding state changes caused by plasma treatment. Figure 7 shows the XPS multi-peak fitting results. After N_2_ plasma treatment, the reduction of Si-O-N bonding was observed. The results show that when the additional nitrogen was incorporated into the nitride layer by the plasma treatment, N vacancies in nitride decreased, suppressing subsequent O interactions. This shows that N_2_ plasma treatment can be effective method to reduce the aforementioned O-related traps that are located at oxide/nitride interface.

## 4. Conclusions

In this paper, N_2_ plasma treatment on silicon nitride is proposed as a solution to suppress the interface trap formation and charge spreading in a SONOS device. In order to investigate the impact of intra-nitride charge spreading on data retention in a 3D SONOS device where the charge trapping layer is shared in a cell string, different gate structures were fabricated using a lithography mask, and the charge loss appeared to be much more severe after baking at 300 °C for 2 h. After N_2_ plasma treatment, both before and after a silicon nitride formation, charge loss was found to decrease. To extract the trap density quantitatively, C–V analysis method was used, which showed an apparent trap decrease, especially in blocking oxide and the trapping layer interface. XPS also showed the reduction of Si-O-N bonding after plasma treatment. The results indicate that N_2_ plasma treatment on silicon nitride is effective to control the shallow O-related interface trap and improve the data retention characteristics of SONOS memory devices.

## Figures and Tables

**Figure 1 micromachines-10-00356-f001:**
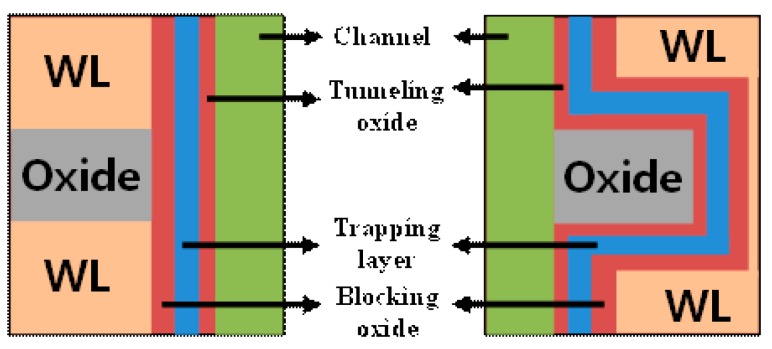
The charge trapping layer structure of (**a**) BiCS 3D NAND and (**b**) TCAT 3D NAND.

**Figure 2 micromachines-10-00356-f002:**
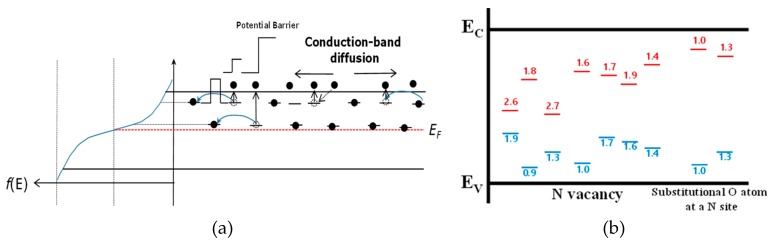
(**a**) Conduction mechanism of programmed Silicon-Oxide-Nitride-Oxide-Silicon (SONOS) memories, (**b**) energy level of silicon nitride.

**Figure 3 micromachines-10-00356-f003:**
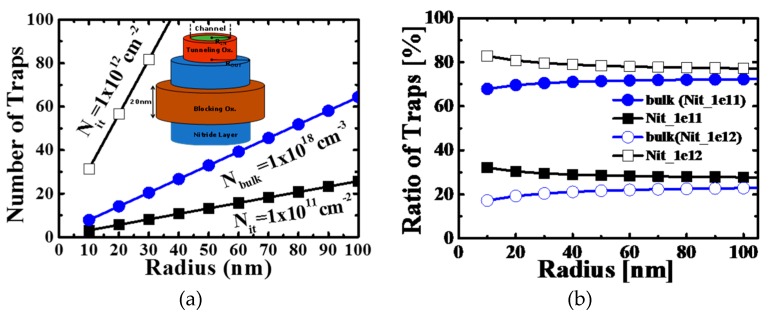
(**a**) Total real number of interfaces and bulk traps and (**b**) the percentage of traps depending on the channel radius of the cylindrical 3D SONOS device. Here, the radius (R_in_ in inset figure) was in the range of 10 to 100 nm, trapping layer thickness was 5 nm and gate length was 20 nm.

**Figure 4 micromachines-10-00356-f004:**
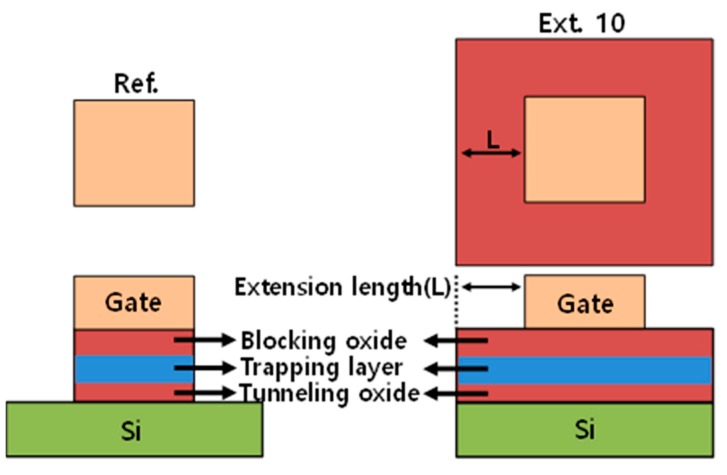
Lithography mask layout to fabricate the test device with a cross-sectional view of the device. Here, Ext. 10 means the extension length of the charge trapping layer was 10 µm. In the case of Ref., the charge trapping layer was etched and self-aligned with the gate and the extension length is 0 µm.

**Figure 5 micromachines-10-00356-f005:**
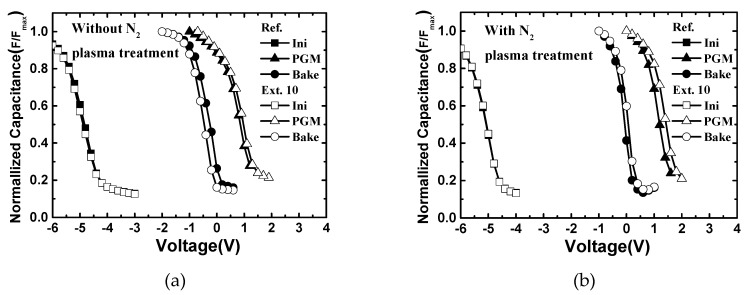
Measurement results of program and data retention characteristics of the fabricated devices (**a**) without N_2_ plasma treatment and (**b**) with treatment. Here, the retention properties were measured after baking at 300 °C for 2 h.

**Figure 6 micromachines-10-00356-f006:**
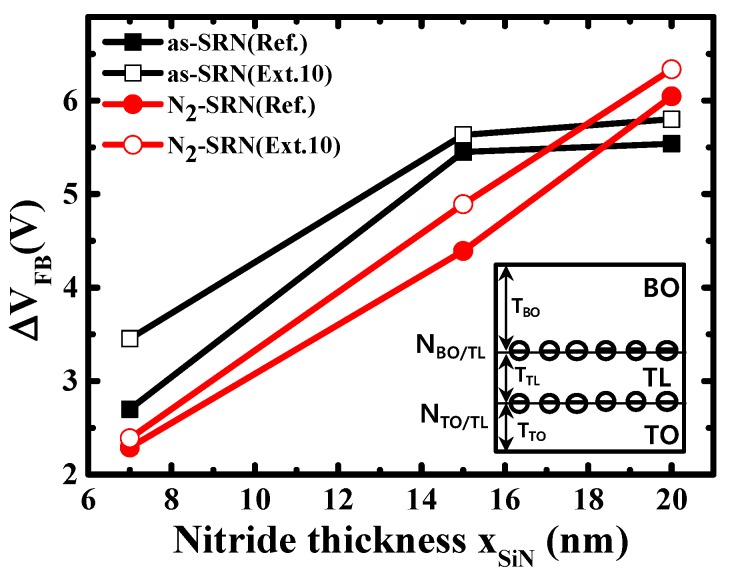
Extracted results of V_FB_ shift in capacitance-voltage curve according to the trapping layer thickness. Inset shows the oxide/trapping layer interface trap sites in the SONOS device structure.

**Figure 7 micromachines-10-00356-f007:**
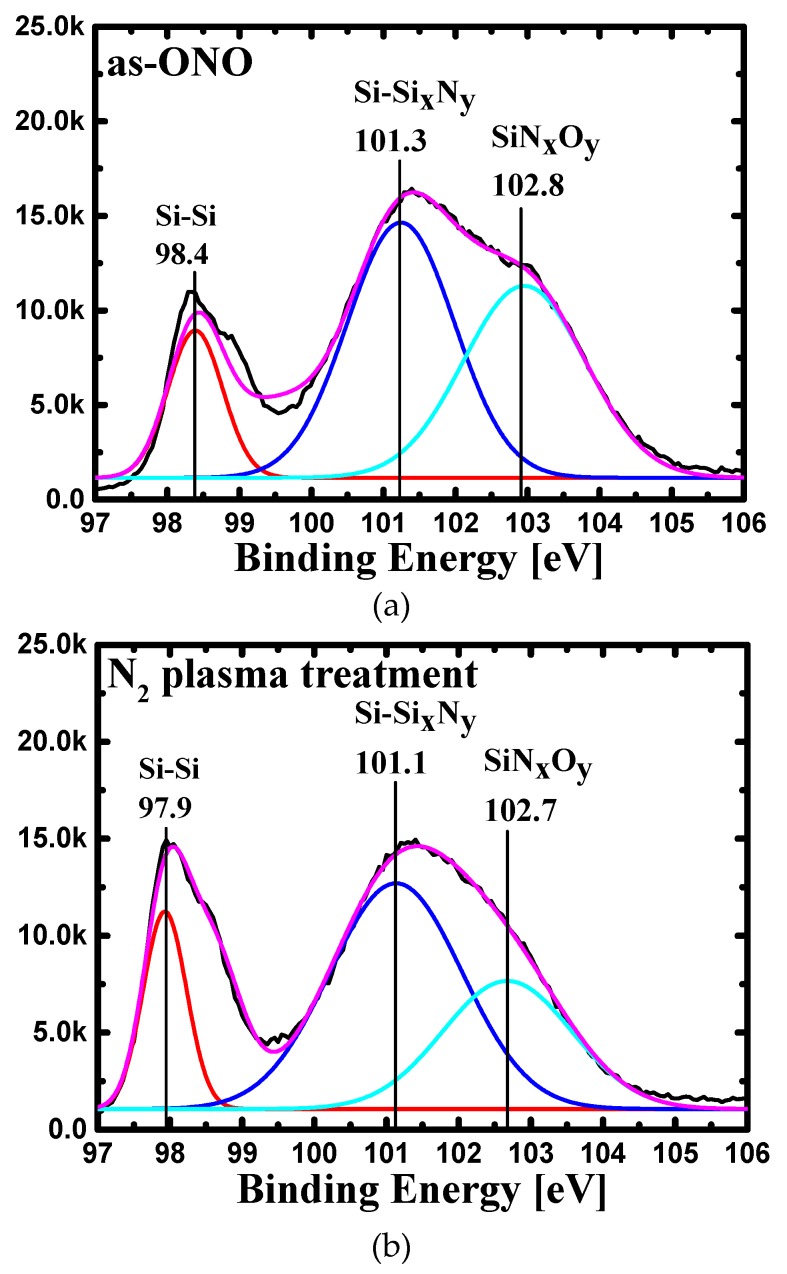
X-ray Photoelectron Spectroscopy (XPS) results of Si2p multi peak fitting of nitride/oxide interface (**a**) as-nitride (without N_2_ plasma treatment) and (**b**) N_2_ plasma treated nitride.

**Table 1 micromachines-10-00356-t001:** Extracted trap density based on C–V analysis. Here, N_BO/TL_ and N_TO/TL_ are the interface trap density of blocking oxide/trapping layer and tunneling oxide/trapping layer, respectively.

Sample	N_BO/TL_ (cm^−2^)	N_TO/TL_ (cm^−2^)	Charge Loss [%]
Ref.	2.53 × 10^12^	8.91 × 10^11^	18.6
Ext.10	4.36 × 10^12^	7.32 × 10^11^	25.7
N_2_ plasma treated Ref.	4.35 × 10^11^	1.11 × 10^12^	17.3
N_2_ plasma treated Ext.10	5.21 × 10^11^	1.18 × 10^12^	20.5

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
