# Peer review of "Investigation of Intra-Nitride Charge Migration Suppression in SONOS Flash Memory"

_micromachines, 2019, doi:10.3390/mi10060356_

Round 1

Reviewer 1 Report

This is a well written article. However, I suggest the following improvements:

Please carefully check equation (1). In the first line of the equation, the terms on the right hand side don’t have the same dimension. It should be noticed that the authors of reference [18] used area densities and volume densities. But in the present article only densities of one type are considered. Therefore, the equations of the cited reference cannot be taken over without changes. Furthermore, line 2 of equation (1) written in bold letters includes densities N_bottom and N_top together with a thickness x_top, which are not explicitly defined in the present article.

I also suggest that the readability of the legend to Figure 6 should be improved.

best regards

The Reviewer

Author Response

Reviewer 1.

This is a well written article. However, I suggest the following improvements:

Please carefully check equation (1). In the first line of the equation, the terms on the right hand side don’t have the same dimension. It should be noticed that the authors of reference [18] used area densities and volume densities. But in the present article only densities of one type are considered. Therefore, the equations of the cited reference cannot be taken over without changes. Furthermore, line 2 of equation (1) written in bold letters includes densities N_bottom and N_top together with a thickness x_top, which are not explicitly defined in the present article.

As you said, we corrected the equation (1) in the previous manuscript with the defined parameters of NTO/TL (interface trap density of tunneling oxide and nitride trapping layer), NBO/TL (interface trap density of blocking oxide and nitride trapping layer), and TTL (thickness of trapping layer), TBO (thickness of blocking oxide), which is expressed as equation (2) in the rewritten manuscript. We also added the original equation in the reference [18] as equation (1) and explained that the equation (1) can be changed to the equation (2) based on the measurement results of DVFB dependency on trapping layer thickness (Fig.6). The corrected ones are highlighted by red letters in the manuscript. (lines 104~120)

I also suggest that the readability of the legend to Figure 6 should be improved.

The legend of Figure 6 is amended with inset which shows the device structure and parameters related with trap and layer thickness like below.

               ( before correction )                  ( after correction )

Reviewer 2 Report

The motivation of the work is relevant, since the treatment or the optimization in the fabrication steps of the devices to improve their performance is critical, especially when they have nanoscale feature sizes. The experimental data supports well what the authors want to demonstrate.

Therefore, I suggest the acceptance of the article in the Micromachines journal. However, I have some questions that I ask kindly the authors for being addressed:

·         Have the authors studied the level of charge loss depending on the plasma treatment conditions? For how long was the N2 plasma applied in the treatments? Is there an influence in the nitrogen plasma treatment time, for example? Could a corresponding experimental graph be presented, at least in a supplementary information file?

·         What was the sampling of the studied devices, both the extended and reference devices? Could some statistics-related data be provided, to know the effectiveness of this treatment in more than singular devices?

·         More details about the C-V measurements set-up is missing. For example: equipment? Under which conditions?

·         Figure 5 should be explained with more detail through the main text.

Also, I would like to suggest some minor improvements:

·         Which is the thickness of the tunnelling silicon oxide formed on the p-type silicon substrate? (lines 63-64)

·         Amend the formula (1) format (lines 97-98)

·         Amend inset of figure 6

·         In figure 5, I believe that the scale of the x axis to represent the graphs should be rearranged. It is difficult to me to observed from the figure the magnitude of the changes before and after N2 plasma. All the curves look like practically superimposed the way they are represented now.

·         With which equipment was performed the XPS spectroscopy?

Author Response

The motivation of the work is relevant, since the treatment or the optimization in the fabrication steps of the devices to improve their performance is critical, especially when they have nanoscale feature sizes. The experimental data supports well what the authors want to demonstrate.

Therefore, I suggest the acceptance of the article in the Micromachines journal. However, I have some questions that I ask kindly the authors for being addressed:

·         Have the authors studied the level of charge loss depending on the plasma treatment conditions? For how long was the N2 plasma applied in the treatments? Is there an influence in the nitrogen plasma treatment time, for example? Could a corresponding experimental graph be presented, at least in a supplementary information file?

I think that your query is very important. I am sorry but the N2 plasma treatment condition is fixed in this experiment. It is mentioned below that we first focused to find out the lateral charge migration (somewhat difficult in large test pattern as you point out!) and then deposition condition of trapping layer, Si3N4. We are continuing this research and just started to change the N2 plasma condition. We will try to find out the details on the plasma treatment.

The time for the treatment is added in the manuscript which is highlighted in red letters. (lines 66~67)

·         What was the sampling of the studied devices, both the extended and reference devices? Could some statistics-related data be provided, to know the effectiveness of this treatment in more than singular devices?

As you know, this experimental procedure (after initial program → erase (the same process with initial UV erase in the industry, program → baking → retention for analysis) needs long time and hard works, and so quantitative statistics-related data was not prepared. Instead, we tried to show that the electrical measurement results on memory performances with electrical trap extraction method and physical analysis method.

We confirmed the lateral migration in the devices with extended charge trapping layer (Figure R1) and N2 plasm treatment effect (Figure R2 and Figure R3). In fact, we are continuing the research on the lateral migration and the suppression process like as N2 plasma treatment. The following results are not published, which shows the effect of the suggested N2 treatment. (The figures are in the attached file)

Fig. R1. Test patterns with different extension length but the same gate area (left) and the measurement results of DVFB according to the test pattern which can be the proof of the lateral migration (right).

Fig. R2. DVFB measurement results according to N2 plasma treatment. Here, the deposition condition of trapping layer is different from the manuscript. We are trying to optimize the Si3N4 deposition condition to improve the memory performance suppressing the lateral charge migration.

Fig. R3. Trap extraction results according to N2 plasma treatment by CV analysis. The bulk trap seems not to be dominant and the N2 treatment is effective to reduce the Ntop (=NBO/TL) interface trap, which is similar with the manuscript.

·         More details about the C-V measurements set-up is missing. For example: equipment? Under which conditions?

More explanation on the C-V equipment and measurement condition is added in the manuscript. (lines 53~54)

·         Figure 5 should be explained with more detail through the main text.

The figure 5 is redrawn and the details on experimental results like as the difference between patterns (Ext. 10 vs. Ref.) and N2 plasma treatment are added, which is highlighted in red letters in the manuscript. (lines 90~99)

Also, I would like to suggest some minor improvements:

·         Which is the thickness of the tunneling silicon oxide formed on the p-type silicon substrate? (lines 63-64)

The tunneling oxide thickness is 6 nm and added in ‘Experiments’ part of the manuscript, which is highlighted by red letters (lines 65~66)

·         Amend the formula (1) format (lines 97-98)

The equation (1) in the previous manuscript is amended and rewritten to equation (2) with the original equation of the reference [18] as equation (1) in this manuscript. (lines 104~120)

·         Amend inset of figure 6

The inset of figure 6 is amended with the defined parameters related with trap and layer thickness.

                ( before correction )                    ( after correction )

·         In figure 5, I believe that the scale of the x axis to represent the graphs should be rearranged. It is difficult to me to observe from the figure the magnitude of the changes before and after N2 plasma. All the curves look like practically superimposed the way they are represented now.

As you said, the figure 6 is difficult to notice the difference between the experimental conditions by superimposed data. We divided the figure 6 into figure 6 (a) without plasma treatment and figure 6 (b) with plasma treatment. In this case, however, the effect of the treatment is not clear even though we extracted and expressed the total charge loss in Table 1. And so, we mentioned the lateral charge loss is estimated to reduce from 28% to 12% by the N2 plasma treatment. The corrected ones are highlighted in red letters in the manuscript. (lines 90 ~99)

                                  (previous Figure. 6)

                                                        (a)                                (b)

                                   (Redrawn Figure. 6)

·         With which equipment was performed the XPS spectroscopy?

Explanation on XPS spectroscopy is added and highlighted in red letters. (lines 55~56).
